# Encapsulation of Black Seed Oil in Alginate Beads as a pH-Sensitive Carrier for Intestine-Targeted Drug Delivery: In Vitro, In Vivo and Ex Vivo Study

**DOI:** 10.3390/pharmaceutics12030219

**Published:** 2020-03-02

**Authors:** Abul Kalam Azad, Sinan Mohammed Abdullah Al-Mahmood, Bappaditya Chatterjee, Wan Mohd Azizi Wan Sulaiman, Tarek Mohamed Elsayed, Abd Almonem Doolaanea

**Affiliations:** 1Advanced Drug Delivery Laboratory, Department of Pharmaceutical Technology, Faculty of Pharmacy, International Islamic University Malaysia, Kuantan 25200, Malaysia; azad2011iium@gmail.com; 2Dentistry College, Al-Kitab University, Kirkuk 36010, Iraq; 3Department of Pharmaceutics, SPPSPTM, SVKM’s NMIMS (Deemed to be University), Mumbai 400056, India; bdpharmaju@gmail.com; 4Naib Canselor, PICOMS International University College, Batu Muda, Batu Caves, Kuala Lumpur 68100, Malaysia; drwanazizi@picoms.edu.my; 5Department of Pharmaceutics, Faculty of Pharmacy, Sinai University, Arish 16020, Egypt; tarek.elsayed@su.edu.eg

**Keywords:** black seed oil, alginate, beads, electrospray, microencapsulation

## Abstract

Black seed oil (BSO) has been used for various therapeutic purposes around the world since ancient eras. It is one of the most prominent oils used in nutraceutical formulations and daily consumption for its significant therapeutic value is common phenomena. The main aim of this study was to develop alginate-BSO beads as a controlled release system designed to control drug release in the gastrointestinal tract (GIT). Electrospray technology facilitates formulation of small and uniform beads with higher diffusion and swelling rates resulting in process performance improvement. The effect of different formulation and process variables was evaluated on the internal and external bead morphology, size, shape, encapsulation efficiency, swelling rate, in vitro drug release, release mechanism, ex vivo mucoadhesive strength and gastrointestinal tract qualitative and quantitative distribution. All the formulated beads showed small sizes of 0.58 ± 0.01 mm (F8) and spherical shape of 0.03 ± 0.00 mm. The coefficient of weight variation (%) ranged from 1.37 (F8) to 3.93 (F5) ng. All formulations (F1–F9) were studied in vitro for release characteristics and swelling behaviour, then the release data were fitted to various equations to determine the exponent (ns), swelling kinetic constant (ks), swelling rate (%/h), correlation coefficient (r^2^) and release kinetic mechanism. The oil encapsulation efficiency was almost complete at 90.13% ± 0.93% in dried beads. The maximum bead swelling rate showed 982.23 (F8, r^2^ = 0.996) in pH 6.8 and the drug release exceeded 90% in simulated gastrointestinal fluid (pH 6.8). Moreover, the beads were well distributed throughout various parts of the intestine. This designed formulation could possibly be advantageous in terms of increased bioavailability and targeted drug delivery to the intestine region and thus may find applications in some diseases like irritable bowel syndrome.

## 1. Introduction

Black seed (Nigella sativa), and specifically black seed oil (BSO), is known to have high therapeutic value. For centuries, it has been used to treat various ailments. Perhaps one of the most significant phytochemical ingredients in BSO is thymoquinone (TQ) [1]. In recent years, researchers have conducted in vivo and in vitro studies to examine the therapeutic value of TQ. These studies have shown that BSO has antihypertensive, antibacterial, antihistaminic, antifungal, analgesic, anti-diabetic, lipid-lowering and anti-inflammatory effects. It can be used as a remedy for symptoms and diseases like asthma, high blood pressure, eczema, cough, headache, influenza, fever, cancer and dizziness [2,3]. Direct addition of oils to the food matrix could cause volatilisation, and subsequently, rapid loss of activity due to exposure to oxygen and ultraviolet light [4]. Moreover, the lack of control of the release rate of the oil produces an instant and short duration effect. One way to avoid these disadvantages is the generation of a polymeric coverage around the oil through microencapsulation or microspherification [5,6].

Microencapsulation is a process where relatively thin coatings are applied to small droplets of liquids and dispersions, or particles of solid material. Microencapsulated oil tends to have a high economic value where certain types of oils, including BSO, are considered as healthy foods and have therapeutic benefits. A microcapsule, or microsphere, protects oils from adverse environmental factors [7,8]. This process has been widely applied to improve efficacy, protect the drug contents, allow sustained-release of tablets, enhance taste masking, lower flavour loss during a product’s shelf-life, extend the flavour perception, sustain mouth-feel over a longer period, and separate chemically incompatible ingredients in a single dosage form [9]. Other benefits of microencapsulation are that it helps sustain absorption, regulate the oil’s release rate, and preserve the appropriate concentrations to produce the desired effect at a specific place [10]. 

The microencapsulation of oils can be challenging whereby sometimes there is a need for proper temperature control to avoid the degradation or the volatilisation of bioactive components. Oils can be encapsulated in several ways. Researchers have widely examined electrohydrodynamic atomisation (EHDA) to produce micro- and nano-sized structures [11,12]. In this regard, there are several factors that need to be considered in order to produce the desired particle size or morphology, including flow rate, applied voltage, nozzle size and the collection distance of the polymeric solution [13]. 

In this study, BSO was encapsulated using alginate, a natural polysaccharide suitable for oral consumption. Alginate is a naturally occurring anionic polymer and can be extracted from brown seaweeds. It is composed of two monomeric structures, 1-4bD-mannuronic (M) acid and a-L-guluronic (G) [14]. It is non-toxic [15] and has high biocompatibility, low cost and high biodegradability [16]. Hence, it is commonly used in the engineering and biotechnological industries. 

Alginate is preferred for encapsulation by ionic gelation as it can form ionic bonds with divalent cations such as Sr^2+^, Ca^2+^, or Zn^2+^, resulting in a cross-linked structure among the G groups of different alginate chains. The egg-box model aptly explains the gelation process where the divalent cation is bounded to two carboxyl groups on the alginate molecules located side-by-side [17]. It is proposed that calcium alginate beads contain hydrogel properties that could control the release of small molecules and macromolecules. The calcium alginate beads also have mucoadhesive properties and could stick to the intestinal mucosa for a long period of time. They also provide a protective structure that could protect oils from environmental factors like acidic media, thus transporting oils into the intestines. Subsequently, studies have used it to facilitate site-specific drug delivery to mucosal tissues [18]. 

In this study, the electro-hydrodynamic technique was adopted to examine the feasibility for encapsulating commercial oils through emulsification. Here, alginate-BSO beads were formulated and characterised in different aspects like yield percentage (Y%), physical appearance (SEM), encapsulation efficiency (EE%), TQ (EE%) (HPLC), particle size distribution, shape, weight uniformity, ex vivo mucoadhesive properties, in vitro drug release profile, and gastrointestinal tract (GIT) distribution. Prior to that, the compatibility was tested using Fourier-transform infrared spectroscopy (ATR-FTIR) and differential scanning calorimetry (DSC).

There are several clinical trials that have reported the therapeutic benefits of BSO related to the gastrointestinal and immune systems. For example, daily twice administration of *Nigella sativa* oil (NSO) (black seed oil) capsules at the dose of 450 mg have reported a potential therapeutic effect on inflammation in the gastrointestinal system [19]. In addition, allergic diseases (allergic rhinitis, bronchial asthma, atopic eczema) were treated with BSO, given in capsules at a dose of 40 to 80 mg/kg/day and 1–2 g/day for 3 months [20]. Furthermore, a dose of 26.7 mg/kg/day of NSO+40 mg omeprazole has a potential synergistic effect on *H. pylori* activity in the gastrointestinal system [21]. Furthermore, direct oral administration of NSO at a dose of 5 mL/day in a honey-based formulation showed significant improvement of dyspepsia severity and *H. pylori* infection in human clinical trials [22]. Another study has reported that NSO at a dose of 500 mg twice/day improved rheumatoid arthritis symptoms as an anti-inflammatory effect [23]. Moreover, it has been reported to have an anti-allergic effect in human studies at a dose of 6 mg/kg of NSO for 30 days oral consumption [24]. Another similar study also reported that oral administration of NSO at a dose of 40–80 mg/kg for 8 weeks showed an anti-allergic effect in 152 patients [25]. However, NSO was administered directly in an oil form in most of the human studies and did not focus on targeted drug delivery due to the lack of formulation development. 

This influences the therapeutic efficacy because the oil may degrade in a stomach environment and lose its bioactivities. Therefore, it was necessary to develop a pH-sensitive BSO carrier to treat gastrointestinal diseases. There is a lack of information about the absorption of the oil or thymoquinone from the oil. There is also a lack of information on the stability of thymoquinone in the oil used in the formulations. This study aimed to provide sufficient information on developing a standard carrier of BSO with assured thymoquinone content. Therefore, this information is highly needed to conduct further in vivo research to optimize the therapeutic human dose, in order to treat various diseases with a standard pharmaceutical dosage form.

## 2. Materials and Methods 

### 2.1. Materials

Sodium alginate with medium viscosity (40–100 mPa s) and high guluronic acid content was obtained from Manugel DMB, ISP, (Sigma-Aldrich, St. Louis, MO, USA). Calcium chloride dihydrate, CaCl_2_·2H_2_O, was purchased from Sigma-Aldrich (Sigma-Aldrich, St. Louis, MO, USA). BSO was purchased from Blessed Seed Sdn. Bhd. (Kuantan, Malaysia). Lecithin was purchased from Fisher Scientific (Fisher Scientific, Waltham, MA, USA).

### 2.2. Preparation of Alginate-BSO Emulsion 

The emulsion was prepared by dispersing BSO (10% *w*/*v* relative to the total emulsion volume) in a solution of 1% *w*/*v* sodium alginate containing lecithin (1, 3 and 5% *w*/*v*). The nano-emulsion was obtained by an ultrasonic homogenizer (QSonica, 53 Church Hill Rd. Newtown, PA, USA) for 55 s at 20% power. 

### 2.3. Characterisation of Alginate-BSO Emulsion

An Olympus light microscope (Olympus, BX53M series, Shinjuku, Tokyo, Japan) was used to determine the emulsion’s microstructure. Subsequently, for determination of particle size distribution and zeta potential, 0.1 mL of the freshly prepared emulsion was diluted with 9.9 mL of distilled water and analysed using a dynamic light scattering instrument (Malvern Zetasizer Nano series Nano-S and Nano-Z, Malvern Instruments Ltd., Worcestershire, UK) [26,27].

To estimate the emulsion stability, the emulsions (*n* = 3) were centrifuged at 4000 rpm for 5 min. The emulsion stability (ES) was calculated according to the phase separation interface position using the following equation:(1)$$\mathrm{ES}\;(\%)\;=\;\frac{V_{\mathrm{emul}}.}{V_{\mathrm{initial}}}\;\times \;100\%,$$
where Vemul is the volume of remaining emulsion after centrifugation and Vinitial is the volume of initial emulsion.

### 2.4. Bead Preparation

BSO alginate beads were prepared using an electrospray technique called electrohydrodynamic atomization (EHDA) [28]. BSO emulsion containing 10% BSO, 1% sodium alginate and 3% lecithin solution was chosen for beads preparation as weight by volume (*w*/*v*). The emulsion was loaded into a 10 mL plastic syringe and pumped through a 22G needle using syringe pump (Shenchen SPLab02, Baoding, China) to control the flow rate. The positive electrode of high voltage power supply (Analog technologies, Inc, San joes, CA, USA) was connected to the needle tip. The collector was a grounded beaker filled with 50 mL of 1% *w*/*w* calcium chloride (gelling bath). The dripping flow rate was varied at 1, 2 and 3 mL/min with the voltage of 3, 5 and 7 kV with 10 cm fixed distance above the surface of the calcium chloride solution. To ensure the complete gelation, the beads were kept in the gelling bath for 30 min for curing with stirring. Then, the beads were separated from the gelling bath using a stainless steel sieve. Then, the collected beads were washed by ultrapure distilled water and after that the beads were dried in a laboratory bench at room temperature for 16 h. The beads yield percentage was determined according to the following equation (Equation (2)):(2)$$\mathrm{Beads}\;\mathrm{yield}\;(\%)\;=\;\frac{\mathrm{Amount}\;\mathrm{of}\;\mathrm{recovered}\;\mathrm{beads}}{\mathrm{Amount}\;\mathrm{of}\;\mathrm{emulsion}\;\mathrm{initially}\;\mathrm{used}}\;\times \;100$$

BSO-free alginate beads, referred to as blank beads, were similarly prepared using 1% *w*/*v* sodium alginate solution containing lecithin 1, 3 and 5% *w*/*v*.

### 2.5. Bead Characterization 

#### 2.5.1. Size and Shape

A Sigma ScanPro 5 image analyser was used to determine the bead size and shape while the images of the wet and dry beads were captured using a digital camera. Here, the roundness of the beads was determined using sphericity factor (SF) [27]. The SF is calculated using the following equation:(3)$$\mathrm{SF}\;=\;\frac{D_{\max}\;-{\;D}_{\mathrm{per}}}{D_{\max}\;+{\;D}_{\mathrm{per}}}$$
where D_max_ is the maximum diameter passing through a bead centroid (mm), and D_per_ is the diameter perpendicular to the D_max_ passing through the bead centroid (mm). A zero SF represents a perfect sphere, while a higher SF indicates a higher shape distortion. Moreover, a bead with SF ≤ 0.05 is considered as a spherical bead.

#### 2.5.2. Encapsulation Efficiency (EE%)

To determine the amount of BSO encapsulated in the beads, the beads were first disintegrated in phosphate buffered saline (PBS). This process returned the beads into emulsion. Then the absorbance was measured at 600 nm using UV-vis spectrophotometer (Shimadzu/UV-1700, Kyoto, Japan) which represents the turbidity of the resulting emulsion. This method is based on previous reports [29,30] that showed a correlation between oil content in the emulsion and the turbidity value. A standard curve was first plotted using a known amount of BSO in the emulsion. The EE was calculated using the equation as follows:(4)$$\mathrm{EE}\;\left(\%\right)=\;\frac{\mathrm{Actual}\;\mathrm{oil}\;\mathrm{content}\;\mathrm{in}\;\mathrm{beads}}{\mathrm{Theoretical}\;\mathrm{oil}\;\mathrm{content}\;\mathrm{in}\;\mathrm{beads}}\;\times \;100$$

#### 2.5.3. Analysis of Bead Weight Uniformity

To evaluate the weight uniformity of the beads, 30 beads were taken and accurately weighed using an analytical balance [31]. The weight of the beads was expressed as mean values of 30 determinations. The coefficient of variations was then calculated.

#### 2.5.4. Scanning Electron Microscopy (SEM)

A Carl Zeiss EvoV R 50 scanning electron microscope (Oberkochen, Germany) was used to observe the microstructure and surface morphology of the BSO alginate beads. Some of the dried beads were cut to examine their inner surface before being spotted onto the aluminium stubs and pasted using carbon adhesive tapes. Subsequently, the beads were sputter-coated with carbon sputter module with the thickness of 100 and 50 Å, respectively, in a vacuum evaporator in an argon atmosphere [18]. Lastly, the coated bead images were taken in a high vacuum condition with accelerating voltage of 10–15 kV. 

#### 2.5.5. Differential Scanning Calorimetry (DSC)

DSC (model 822, Mettler Toledo, OH, USA) was used to examine the compatibility and thermal characteristics of BSO-loaded and BSO-free alginate beads. About 3.20 mg of beads were weighed and sealed in a standard aluminium pan. The samples were analysed under a temperature range from 50–350 °C and heated under a stream of nitrogen flow rate of 20 L/min at a heating rate of 10 °C/min [32]. 

#### 2.5.6. ATR-FTIR 

ATR-FTIR spectra of sodium alginate, BSO, and beads were recorded between 4000–400 cm^−1^, with a resolution of 1 cm^−1^ [32] with ambient air as a background using PerkinElmer Spectrum 100 spectrophotometer (Perkin Elmer Corp., Norwalk, CT, USA). The spectra were processed using the SpectraGryph 1.2 spectroscopy software.

#### 2.5.7. Determination of Thymoquinone 

A 100 µg/mL TQ standard stock solution was prepared by dissolving 5 mg of TQ in 50 mL of methanol. The serial dilution of TQ stock solution was performed to get different concentrations up to 20 µg/mL. The solution was then filtered using a 0.2 µm syringe filter and 2 mL of the solution was placed in HPLC vials for further analysis. 

TQ was first quantified in the BSO used to prepare the beads. Fifty milligrams of BSO was carefully weighed then 10 mL methanol was added and mixed thoroughly using a vortex mixer and centrifuged at 4000 rpm for 5 min. Then, the upper layer (methanol) containing the TQ was collected and filtered using a 0.2 µm syringe filter. For the extraction of TQ from alginate beads, 0.3 g of dried alginate-BSO beads was broken by adding 1 mL of mixture of 0.5 M EDTA, phosphate buffer solution, pH of 6.8, with a volume ratio of 1:1. Subsequently, the mixture was placed for 30 min in an incubator shaker at 150 rpm. After the broken beads were completely transformed into an emulsion, 10 mL of methanol was added and mixed thoroughly with the emulsion. The mixture was immediately centrifuged at 4000 rpm for 5 min. The supernatant layer (methanol) was then collected and filtered using a 0.2 µm syringe filter.

A HPLC method was used to quantify TQ as described in a previous study [33]. The analysis was performed using the HPLC system (Shimadzu LC-20AT, Shimadzu, Japan) at a column oven temperature of 40 °C and a sample cooler temperature of 8 ± 0.2 °C. Acetonitrile and water mixture with the volume ratio of 60:40 was used as the mobile phase, with a flow rate of 1 mL/min through an Inspire C18 analytical column (4.6 × 250 mm, 5 µm). Meanwhile, TQ detection was conducted using a diode-array detector at a wavelength of 254 nm and each sample’s injection volume and run time were predetermined as 20 µL and 10 min, respectively. 

#### 2.5.8. Swelling Characteristics of Beads

The swelling behaviour of the optimised BSO-loaded bead formulations (F1–F9) was measured according to the weight gain percentage in two different aqueous media: the simulated gastric fluid (SGF, 0.1 N HCl, pH 1.2) and simulated intestinal fluid (SIF, pH 6.8). The conditions were maintained for 2 h at 37 ± 0.5 °C while 100 mg of dried alginate-BSO beads was immersed in 50 mL of both SGF and SIF fluids media [34]. 

At each predetermined time interval (0, 10, 20, 30, 40, 50, 60, 70, 80, 90, 100, 110 and 120 min), the swollen beads were removed and filtered using a metal mesh. The swollen beads were dried with a paper towel to eliminate the excess fluid. The following equation was used to determine the percentage of swelling index,
(5)$$\mathrm{Swelling}\;\mathrm{Index}=\frac{W_{t}-W_{o}}{W_{o}}\times 100$$
where W_o_ is the dried beads’ initial weight and Wt is the weight of swollen beads at time t.

In addition, pictures of the beads were taken using a digital camera at the same time intervals after the weights of the swollen beads were measured. The bead samples were then characterised based on their thickness and morphological appearance. 

#### 2.5.9. In Vitro Drug Release 

The dissolution was carried out at 37 ± 1 °C and 50 rpm using an orbital incubator shaker under continuous stirring (Innova™ 4000 Benchtop Orbital Shakers, New Brunswick Scientific™, Edison, NJ, USA). BSO beads (0.1 g) were added to 200 mL of SGF (pH 1.2) and the test was carried out for 2 h, and afterwards it was carried out in SIF (pH 6.8) for another 2 h. Five mL of aliquots were collected at regular time intervals and were replaced with a similar amount of fresh dissolution medium. The collected aliquots were filtered, suitably diluted using the dissolution medium, and the absorbance was measured with a UV–vis spectrophotometer (Shimadzu/UV-1700, Kyoto, Japan) at 600 nm.

The in vitro drug release data were then fitted to mathematical models like Korsmeyer–Peppas, zero order, first order, Higuchi, and Hixson–Crowell, to identify the drug release mechanism [34]. 

These models were compared by calculating the squared correlation coefficient (r^2^) value [35]. Meanwhile, the Korsmeyer–Peppas model was employed to identify the in vitro drug release mechanism of the formulations. This process was done to pin-point competing release mechanisms, including case-II transport (relaxation-controlled release), Fickian release (diffusion-controlled release) and non-Fickian release (anomalous transport). The Fickian and non-Fickian release was determined by the value of n where Fickian release is evident when n is 0.5 while non-Fickian release occurs when 0.5 < *n* < 1.0. Besides that, *n* = 1.0 indicates case II transport and *n* > 1 indicates super case II transport [36].

#### 2.5.10. Ex Vivo Mucoadhesion Test 

The formulation’s mucoadhesive properties were evaluated using the modified wash-off method [35]. A fluorescence dye (6-coumarin) was added to the beads. After being sacrificed, the rats were immediately dissected where their intestines were removed, washed with normal saline and subdivided into three 14 cm sections. One hundred beads were scattered uniformly on the intestinal mucosa surface and the intestine loaded with the beads was set up on an aluminium sheet which was fixated at 45° in a horizontal plane. Two different wash-off media, acid media (pH 1.2) and phosphate buffer (pH 6.8), were warmed to 37 ± 1 °C and passed at a rate of 5 mL/min over the intestinal section up to 6 h. Then, the visualization of the fluorescent beads remained at the intestinal mucosa surface, was assisted by UV light illumination, then counted at 1 h time intervals up to 6 h. The percentage of remaining beads was calculated, and the data were statistically analysed using ANOVA. Visual inspection was used to record the time needed to detach the beads from the rat intestine mucosal surface. 

#### 2.5.11. Gastrointestinal Tract (GIT) Distribution 

The study used Sprague–Dawley (SD) rats and the protocol for these tests has been approved from IIUM animal ethics committee dated 14/05/2018 (Approval No. IIUM/504/142/IACUC-2018(8)). Twelve rats were included in this study and all were between 6–8 weeks old and weighed between 200–220 g. The rats were not provided any food for 20 h before the experiment, except normal tap water that was provided ad libitum. The rats were divided into 3 groups with 4 rats in each. A fluorescent dye (6-Coumarin) was added to prepare the microsphere beads. Each rat was orally given 100 beads suspended in distilled water using 16G oral gavage under non-anaesthetic conditions before they were sacrificed at 1, 3 and 6 h post administration. After being sacrificed, the rats were immediately dissected where their stomach and intestine were removed and cut into pieces. The stomach was opened and washed with normal saline while the intestine was subdivided into six 14 cm sections (labelled as Sections 2–7, where section 1 is the stomach) [37]. These sections were carefully cut and opened to reveal the inner mucosal surface. The beads used in this study were added to a small amount of coumarin-6 as a fluorescent marker to facilitate the tracking in the GIT. The sections were put under a UV light to calculate the beads found in each section of the intestine and the stomach, with the aid of a mobile phone camera and a commercial green camera filter. Lastly, the percentage of beads found in the intestine and stomach was calculated before being statistically analysed using ANOVA.

### 2.6. Statistical Analysis

Two software programs, Minitab and Design-Expert 8.0.6.1 software (Stat-Ease Inc., Minneapolis, MN, USA) were used to perform the statistical optimisation, while the KinetDS 3.0 Rev. 2010 software was used to ascertain the squared correlation coefficients (r^2^) of all kinetic models. Each measurement was done in triplicate (*n* = 3) and the measured data are expressed as the mean ± standard deviation (SD). Data were compared by *t*-test or one-way ANOVA followed by Tukey’s test considering *p* < 0.05 as indication of significant difference. 

## 3. Results and Discussion

### 3.1. Alginate-BSO Emulsion

Lecithin concentration had significant influence on BSO emulsion stability. Emulsions prepared form all 3 lecithin concentrations were relatively stable. BSO emulsion with 5% *w*/*v* lecithin concentration exhibited the maximum emulsion stability followed by 3% and 1%. As lecithin concentration increased, the emulsion particle size became larger as shown in Table 1. This could be related to the increased viscosity at higher lecithin concentrations. A more viscose continuous phase can limit the effectiveness of size reduction using the ultrasonic homogenizer. In addition, a thicker layer of the surfactant (lecithin) adsorbs on the oil droplets at higher lecithin concentrations, leading to larger particles. The microscopic view of the emulsion did not change dramatically before and after centrifugation (Figure 1), confirming the relatively good stability of the emulsions.

Lecithin concentration has an influence on the distribution of particle size and zeta potential [38]. The highest polydispersity index (PdI) was 0.422 ± 0.04 (before centrifugation) and 0.548 ± 0.11 (after centrifugation) (*p* ˂ 0.05) and was observed with 5% *w*/*v* lecithin concentration. Since a PDI value of 0.300 and below indicates homogenous particle size distribution [35], the emulsions showed a non-homogenous particle size distribution as shown in Figure 2. This might be due to the sonication procedure which relies on manual operation. Automating this step with continuous mixing during the sonication may improve the polydispersity of the emulsion. This can be optimized upon scaling up the preparation. 

Zeta potential has a direct effect on the stability of the colloidal structure. The emulsion droplets practically had high negative charges due to the chain of alginate being also negatively charged because of the presence of hydroxyl and carboxyl groups in the structure of the alginate molecule [39]. In short, before and after centrifugation, a considerable change was not observed on the emulsion’s particle size, PDI and zeta potential.

### 3.2. Percentage of Yield (Y%)

On average, the percentage of yield was 94.87 ± 2.11% (Table 2). The loss in the beads can be attributed to the dissipation of emulsion during the emulsification process. In addition, some amount is probably lost during the electrospray process through the needle. A previous study reported an 80% yield of encapsulation of limonella essential oil using the solvent evaporation technique [6]. Moreover, another study reported a very low yield (62.3%) for thyme essential oil loaded alginate beads. The encapsulation was performed using ionic gelation technique without applying electric voltage [1]. In this experiment, the obtained yield is high compared to previous studies. 

### 3.3. Characterization of Beads

#### 3.3.1. Size and Shape

The size of the wet beads ranged from 4.76 ± 0.01 mm (F3) to 1.50 ± 0.02 mm (F8) and dry beads were from 1.86 ± 0.02 mm (F2 & F3) to 0.58 ± 0.01 mm (F8) (Table 2). Few factors influenced the size and sphericity of the beads. The electrical voltage and emulsion flow rate were notable factors that affected bead size and sphericity. When increasing the flow rate, the bead size and sphericity factor also increase. On the other hand, when increasing the electrical voltage, the bead size and sphericity factor decrease. However, if the electrical voltage and flow rate are increased proportionally, the bead size and sphericity factor do not decrease at a specific point (Figure 3A,B). A flow rate of 2 mL/min and an electrical voltage of 7 kV was found to produce the desired bead size and a spherical shape (Table 2). The desired size was set according to the inner size of the oral gavage ball diameter used in the animal administration. The inner size of the G16 gavage is 1.19 mm and our obtained size is 0.58 ± 0.01 mm.

#### 3.3.2. Encapsulation Efficiency

The percentage of EE of wet and dry bead formulations ranged from 67.20 ± 1.46% (F1) to 104.50 ± 4.04% (F8) and 59.92 ± 0.90% (F1) to 90.13 ± 0.93% (F8), respectively (Table 2). The overall results showed that EE% increased with the increase in voltage and flow rate. This may be due to high voltage which may accelerate the affinity of cross-linking with CaCl2 gelling solution via high numbers of positions for ionic cross-linking to entrap a high amount of oil and prevent oil leakage from the polymer matrix.

One of the main objectives of this experiment was to achieve the targeted size, sphericity factor, EE% and drug release of BSO-loaded beads. In this sense, high electrical voltage (kV) was employed to achieve the goals. However, to adjust and optimize the emulsion flow rate along with a high electrical voltage was the big challenge. To optimize the formulation, different flow rates (1, 2, 3 mL/min) and voltages (3, 5, 7 kV) were applied. It is well understood from this experiment that flow rate and voltage have a significant impact on the size of BSO-loaded beads. The main effect clearly showed that as the voltage increases, bead size decreases from 1.86 ± 0.02 mm to 0.58 ± 0.01 mm. However, the bead size increased from 0.72 ± 0.01 0.93 ± 0.01 mm when the flow rate increased from 1 mL/min to 3 mL/min at 7 kV. The formulation (F8) achieved the lowest size of 0.58 ± 0.01 mm with the flow rate of 2 mL/min and voltage 7 kV (Figure 3A). Moreover, medium voltage (5 kV) and high flow rate (3 mL/min) generated a more spherical shape of beads compared to low and medium flow rate (1, 3 mL/min) and low and high voltage (3, 7 kV) (Figure 3B). Remarkably, high voltage (7 kV) and medium flow rate (2 mL/min) showed the highest encapsulation efficiency (˃90% and ˃80% for wet and dry beads, respectively) (Figure 3C). Regarding drug release, high voltage and medium flow rate exhibited the maximum cumulative percentage of drug release in alkaline media (Figure 3D). The interaction effect plots showed the effect of interaction between voltage and flow rate and their influence on size, sphericity factor, encapsulation efficiency and drug release of BSO-loaded beads (Figure 4). 

Piornos et al. (2017), reported that the EE depended on the strength of the encapsulating matrix [40]. Moreover, the cross-linking ability on the outer surface of alginate beads is another factor to increase the EE of oil in alginate. They have found the highest EE (98.30%) of linseed oil with the optimal oven-dried alginate beads where the size of the beads was from 1.90 to 2.94 mm and sphericity factor was 0.044–0.168. [41]. However, the percentage of drug delivery into the targeted site was only 83% in digestive conditions. Another study also reported similar findings, with 98.7% EE, where the size was 2.13 mm and sphericity factor was 0.08 mm [42]. On the other hand, the dry electrosprayed BSO-loaded alginate beads in our study showed EE of 90.13 ± 0.93% with more than 90% release in the targeted SIF as shown in the release study. This indicates that alginate beads were able to successfully encapsulate BSO and release it almost completely into the targeted site.

#### 3.3.3. Analysis of Beads Weight Uniformity

In any pharmaceutical dosage formulation, it is necessary to maintain good manufacturing practices for the manufacturer. Its dose serves as an indicator of the quality of dosage formulation as well as quantity of active pharmaceutical ingredient (API) filled or encapsulate in the formulation. The coefficients of weight variation of BSO-loaded alginate beads varied from 1.37% to 3.93%. The recent findings exhibited that the mean weight of all the BSO-loaded beads did not deviate more than 5% (Table 2). According to USP pharmacopeia specifications, all these formulations (F1–F9) met the weight uniformity criteria [43]. The weight uniformity of these beads indicates the uniform encapsulation of the BSO using electrohydrodynamic-assisted alginate beads.

#### 3.3.4. ATR-FTIR 

ATR-FTIR spectra of sodium alginate, lecithin, CaCl2, BSO, alginate-BSO beads, blank beads, and physical mixture are shown in Figure 5. The characteristic bands of sodium alginate appeared at 3400 cm^−1^ for O–H stretching and at 1595 and 1408 cm^−1^ for C=O and C–C groups resulting from stretching vibration. A previous study also reported separate bands at 1600 and 1405 cm^−1^. These two bands were attributed to asymmetric and symmetric stretching vibration of –COOH groups [28]. Moreover, the bands in the sodium alginate spectra at 1316 and 1025 cm^−1^ are due to the C–H group. The lecithin spectra showed three bands at 2922, 2853 and 1465 cm^−1^ which are attributed to the C=O, C–H and C–C groups, respectively. The noticeable peaks for BSO were at 2953 and 2853 cm^−1^ (C–H in CH_2_ and C=C–H respectively). Moreover, the bands that appeared at 1753, 1653 and 1453 cm^−1^ indicated C=O, C=C and C–H, respectively due to predominance of carbon chains in the fatty acids. In general, fatty acids compose more than 98% of BSO. The present findings are very similar to previous reports [30,43]. In calcium chloride spectrum, the band at 1613 cm^−1^ is due to the symmetrical stretching bond of Ca–Cl. The broad peak at 3441 cm^−1^ is because of the O–H stretching vibration, probably originating from the residual water.

The physical mixture spectrum of alginate, lecithin, CaCl2 and BSO seems to be an overlay of the individual spectra whereby the peaks appeared in their respective locations. For example, the peak from sodium alginate at 1595 and 1408 cm^−1^ remained in the physical mixture. Similar observation was noticed for the peaks of 1753 and 1408 cm^−1^ of lecithin. In the alginate-BSO bead spectra, most of the peaks in BSO were not seen, which indicates the encapsulation of BSO inside the beads and absence of the oil on the bead surface. In addition, the peaks of the carboxyl group in sodium alginate shifted to higher values, from 3400 to 3424 cm^−1^ in the blank and BSO-loaded alginate beads [44]. This is attributed to the complex formation with calcium ions. The absence of the Ca–Cl band at 1613 cm^−1^ also confirms the complex formation. The above results confirm the complexation between sodium alginate and calcium chloride and the encapsulation of BSO into the alginate bead nanocavities. This complex occurred only in the solution but not in the dry physical mixture.

#### 3.3.5. DSC

DSC is employed for the analysis of thermal behaviour of the active drug molecules and the excipients of the formulation, as well as their interactions throughout the preparation development process [42,43]. In this study, DSC provides the thermal profile of BSO-free (blank) and BSO-loaded beads. BSO-free beads showed a broad endothermic peak at 86.7 °C attributed to the loss of water from the bead gel structure, and a sharp endothermic peak at 203 °C due to glass transition of the gel [5]. Moreover, BSO-free beads showed two exothermic peaks at 200 °C and 275 °C, respectively (Figure 6a). They result from degradation of polyelectrolytes due to dehydration and depolymerization reactions most probably due to the partial decarboxilation of the protonated carboxylic groups and oxidation reactions of the polyelectrolytes.

On the other hand, the above-mentioned thermal events disappeared in the DSC thermogram of BSO-loaded beads. This might be attributed to the lower moisture content since the oil is dispersed in the bead matrix, making it less hygroscopic. The low moisture content and the absence of interaction between the oil and the gel matrix might also increase the stability of the beads whereby the degradation peaks were not observed within the test temperature and time. 

#### 3.3.6. SEM

The scanning electron microphotographs of the optimized BSO-loaded dried beads revealed the surface and internal structure containing a large number of pores or channels in the beads (Figure 7). The pores and channels were probably formed as a result of the transfer of water molecules in the polymer network during the drying process.

#### 3.3.7. Quantification of TQ 

The HPLC chromatograms showed well resolved peaks of TQ with no interference. Chromatograms of the blank, standard TQ, BSO-loaded beads and BSO are shown in Figure 8. It has a retention time (Rt) of 7.060 min at 254 nm. Similarly, BSO-loaded alginate beads and BSO alone also shown Rt of 7.063 and 7.061 min at the same wavelength, respectively. The percentage of TQ in BSO before encapsulation was 1.96 ± 0.01% (*w*/*w*). However, the percentage of TQ in the oil of BSO-loaded alginate dried beads was 1.589 ± 0.158% (*w*/*w*). This means that the EE of TQ in the optimized BSO-loaded alginate beads (F8) is 81.07 ± 0.23%. This highlights that even though the EE of the oil was 90.13 ± 0.93%, some TQ from the oil degraded or leaked to the gelling bath. This is attributed to the solubility of the small amount of TQ in water and the low stability of TQ in aqueous solutions. 

#### 3.3.8. Bead Swelling Behaviour 

Bead swelling is a significant behaviour that influences the drug release pattern of polymeric materials [45]. The swelling behaviour of the beads containing BSO was evaluated in simulated gastric fluid (SGF, 0.1 N HCl, pH 1.2) and simulated intestinal fluid (SIF, pH 6.8). The data are presented in Figure 9.

The swelling parameters such as swelling exponent (ns), kinetic constant (ks), swelling rate (%) and water penetration velocity (up to 1 h) were determined to understand the swelling mechanism (Table 3). The percentage swelling index of all bead formulations (F1–F9) were at first lower in SGF, in contrast with that of SIF, demonstrating a pH-sensitive swelling behaviour. This might have occurred due to shrinkage of alginate at acidic pH [46]. Therefore, higher percentage of swelling index of these beads were noticed at 30–90 min in SIF (pH 6.8). The exchange of calcium ions can be used to explain the swelling behaviour of the beads in alkaline pH where the presence of calcium-sequestrant phosphate ions helps bind the carboxylic groups of cross-linked alginate lecithin matrix and the sodium ions found in phosphate buffer. This causes turbidity in the phosphate buffer as calcium phosphate is formed [47]. Studies have found that alginate has hydrophilic properties. In this light, while it has limited swelling characteristics in acidic pH, it will swell in neutral pH conditions in the intestine or the colon [48]. Subsequently, the alginate beads will disintegrate as the calcium ions in the egg-box buckled structure are diffused out and enter the swelling medium. The same ion exchange can also be observed in the calcium-coordinated carboxylic groups found in the alginate. This finding showed that optimized calcium alginate BSO beads slightly swell in the acidic pH of the stomach, and subsequently, the beads swell more as they move towards the more alkaline conditions of the upper intestine. 

The hydration of the alginate hydrophilic groups causes the dry beads to swell, due to differences in osmotic pressure of the fluid inside and outside the beads [49]. The swelling mechanism used here is closely linked to the Ca^2+^ and Na^+^ reversible ion-exchange reaction [50,51,52]. Recently, Bae et al. (2019) reported the swelling ratio of alginate microbeads was abruptly increased in the presence of Na^+^ ions reached in sea water media [53]. Here, larger swelling was observed as water freely penetrates the beads and fills the inert pores in the polymer chains. The beads showed larger swelling as they are exposed to the alkaline phosphate buffer pH 6.8 environment and the swollen beads would eventually release the entrapped material and disintegrate. 

Moreover, erosion and dissolution of swollen cross-linked BSO-loaded alginate beads occurred within 100–110 min. All formulations were completely eroded at 120 min. One of the main purposes of this research is to release the drug in the intestine where drugs are usually absorbed. In acidic pH 1.2 (SGF), all formulations (F1–F9) retained their original shape or shrank, which reduces their disintegration tendency (Figure 10). The main structure of the beads is due to the formation of the Alg-Ca^2+^ complex alginate. The presence of carboxyl groups in the alginate network helps to retain the beads without swelling.

Oppositely, in SIF, BSO-loaded alginate beads rapidly expanded (Figure 11). All formulations (F1–F9) were entirely dissolved the SIF (pH 6.8) at 120 min (Figure 11, T120). Calcium alginate complex solubility was increasing in higher pH conditions due to the presence of phosphate ions in the medium, which have more affinity to Ca^2+^ than alginate ions. As a result, alginate-calcium was decoupling and the gel dissolved in the phosphate buffer. 

These BSO-loaded alginate beads can effectively shield the encapsulated therapeutic ingredients from the harsh-lytic condition of the stomach throughout its travel through the gastrointestinal tract. Therefore, these formulations can favourably deliver the drug to the specific targeted site, such as the intestine or colon, by the influence of the pH environment. 

#### 3.3.9. In Vitro Drug Release 

The drug release from alginate beads relies on the dissolution medium penetration into alginate beads, swelling and dissolution of the alginate hydrogel, and the dissolution of the encapsulated drug subsequent to leakage through the swollen hydrogel. The drug release pattern of BSO-loaded wet and dried beads in SGF (pH 1.2) and SIF (pH 6.8) at 37 °C is shown in Figure 12 and Figure 13. The data showed that the percentage and rate of drug release increased in the SIF. This is consistent with the swelling character of BSO-loaded beads which outlines that the swelling index percentage of the beads increased in the SIF. This higher release rate could be attributed to the electrostatic repulsion among negatively charged carboxyl groups of alginates in the beads [54,55].

The amount of drug released from BSO-loaded beads was less than 5% in SGF and more than 90% in SIF at 2 h. This finding suggests that BSO-loaded beads can be used as a useful drug carrier for specific-site targeted drug delivery, such as intestine/colon-targeted drug delivery, without premature drug release during its transition period in the stomach.

According to Corstens et al. (2017), the physical characteristics of the alginate beads are largely influenced by the pH and ionic concentrations [56]. In this regard, while the alginate beads showed excellent gastric stability, they swelled and disintegrated in simulated intestinal conditions [51]. This is supported by other studies that reported that the beads shrink under gastric conditions, before swelling and disintegrating at the end of the intestinal phase [57,58]. Studies also found that across the beads, the gel structure is not necessarily homogeneous, as under intestinal conditions, alginate beads tend to have larger pores in their core [59]. The release profile demonstrated that compared to the simulated intestinal fluid (pH 6.8), the cumulative percentage of oil released in the simulated gastric fluid (pH 1.2) is significantly lower (*p* < 0.05). The low percentage of oil release in this media could be due to the absence of any potential swelling rate in the simulated gastric fluid. In the meantime, the study observed a positive correlation between the erosion rate and the release of oil. 

More oil was released from the beads as the erosion rate in the intestinal phase increased [60]. It was also observed that while the sodium alginate has a structural resistance to the acidic environment, it was rapidly released in the mild alkali condition [61]. The present findings demonstrate that the BSO-loaded beads can be used for intestinal site-specific drug delivery carriers demonstrating pH-dependent drug release behaviour as they travel across the GI track.

#### 3.3.10. Kinetics Modelling and Mechanism of Drug Release

To better comprehend the mechanism of BSO release from electrohydrodynamic-assisted alginate beads, the cumulative percentage of drug release data was fitted into different equation models to obtain the correlation coefficient (r^2^) and release exponent (*n*) values (Table 4). The correlation coefficient (r^2^) value of BSO-loaded beads (F1–F9) in SGF were found to fit first order kinetic model (r^2^ = 0.818–0.981) over the predetermined release time of 2 h. However, r^2^ data for all the formulations (F1–F9) in SIF followed zero order (r^2^ = 0.926–0.980) and Korsemeyer–Peppas (r^2^ = 0.900–0.997) kinetic model at 2 h. The diffusional or release exponent (*n*) values of the formulations (F1, F2, F4) ranged from 0.49 to 0.61, indicating non-Fickian diffusion or anomalous transport in acidic media (pH 1.2). This is regarded as two different mechanisms such as couple diffusion or followed by polymer relaxation (erosion), whereas the rest of formulations followed the super case II transport model. On the other hand, the release exponent value for all the formulations (F1–F9) in SIF were ˃0.85 (0.86–1.66), indicating that the BSO release from alginate beads followed super case II transport mechanism. This release mechanism occurred through the polymer dissolution controlled and polymeric chain expansion or polymer relaxation/swelling. This is in line with the swelling study which showed the same pattern in release behaviour mechanism.

#### 3.3.11. Ex Vivo Mucoadhesion Test of Beads

The ex vivo wash-off time revealed that the beads could adhere to the mucosal surface of the intestine up to the experimental period. In this part, data were acquired according to predetermined time interval from 1 to 6 h with two different wash-off media. Firstly, in SIF the beads showed a good mucoadhesive strength whereby the percentage of remaining beads was recovered from the mucosal surface of the employed intestine as 98.26 ± 1.15% at 1 h and 38.83 ± 2.51% at 6 h. This practically supports its potential mucoadhesive properties in intestinal media (pH 6.8). Secondly, the mucoadhesive properties in SGI showed very weak mucoadhesive strength (65.42%) (Figure 14). The presence of –OH groups in alginate could confer the mucoadhesive property in these beads as it forms hydrogen bonds with the mucus molecules. In this light, the hydroxyl groups of the hydrophilic polymers attach themselves to the mucus membrane via different methods such as van der Waal’s forces, hydrogen bonding, and ionic interactions [62]. This reflects that polymer network matrices influence the mucoadhesiveness of the alginate beads. This study also observed that the newly optimised formulation contains a higher proportion of polymer compared to other formulation. In the meantime, there is strong electrostatic attraction leading to good mucoadhesion between mucin and polymer (alginate) matrices. Wu et al. (2019) reported the ex-vivo mucoadhesion properties of the sodium alginate microspheres which was retained 88.59 ± 2.17% in rat gastric mucosa [63]. On the other hand, this current ex vivo study showed that 98.26 ± 1.15% was retained of BSO-loaded alginate beads after 1 h in rat intestinal mucosa. 

#### 3.3.12. GIT Distribution Pattern of BSO-Loaded Alginate Beads

Following oral administration of fixed number of beads, more than 90 ± 4.62% beads were recovered from the stomach after 1 h, but at the same time, less than 10% (7.0 ± 0.75%) were in the duodenum followed by 3% in the jejunum. However, at a certain time interval (3 h), the maximum number of beads were transferred or moved from the stomach to the duodenum (14 ± 1.21%) and the lower part of intestine, including the major sections of jejunum-1 (23 ± 2.33%), jejunum-2 (21 ± 1.98%), jejunum-3 (23 ± 2%), ileum (11 ± 1%) and colon (6 ± 0.54%). Furthermore, after 6 h of administration, significantly, a large quantity of beads (6%) was counted from different parts of the jejunum (3, 4, 5) and ileum. After 1 h of oral administration, almost all beads were retained in the stomach, while beads were not observed in the rest of the intestine sections except the duodenum. After 3 h of beads administration, the maximum number of beads in the stomach moved to the duodenum and jejunum regions of the lower intestine which was exposed to UV fluorescence. Moreover, various sections of the intestine at 6 h of administration were not noticed. Furthermore, mucoadhesive and swelling characteristics of the beads in intestinal fluid facilitated the erosion of the polymer matrix and the drug was released in the specific site of the intestine in the next 2 h. Due to erosion, beads were not seen in the lower sections of the intestine at 6 h of administration. In the distribution study, qualitative image analysis showed the uniform distribution of beads at 3 h of oral administration in rats.

## 4. Conclusions

Novel BSO-loaded alginate beads were prepared as a pH-sensitive carrier for intestinal site-specific drug delivery. It showed pH-dependent swelling and drug release behaviour. The swelling of BSO-loaded beads was less in SGF compared to SIF, which can be advantageous as an intestinal-specific drug carrier. The release of BSO from the BSO-loaded beads was dependent on pH. The rate of BSO released from the beads was higher in SIF due to the extended swelling of the beads. These results indicate that BSO-loaded beads could be an oral drug delivery system to deliver drugs more specifically to the intestine.

## Figures and Tables

**Figure 1 pharmaceutics-12-00219-f001:**
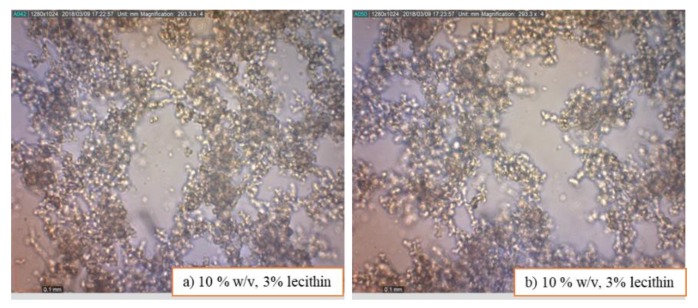
Microscopic image of oil droplets in black seed oil (BSO) emulsion. (**a**) Freshly prepared and (**b**) after centrifugation at 4000 rpm for 5 min. Magnification at 10×.

**Figure 2 pharmaceutics-12-00219-f002:**
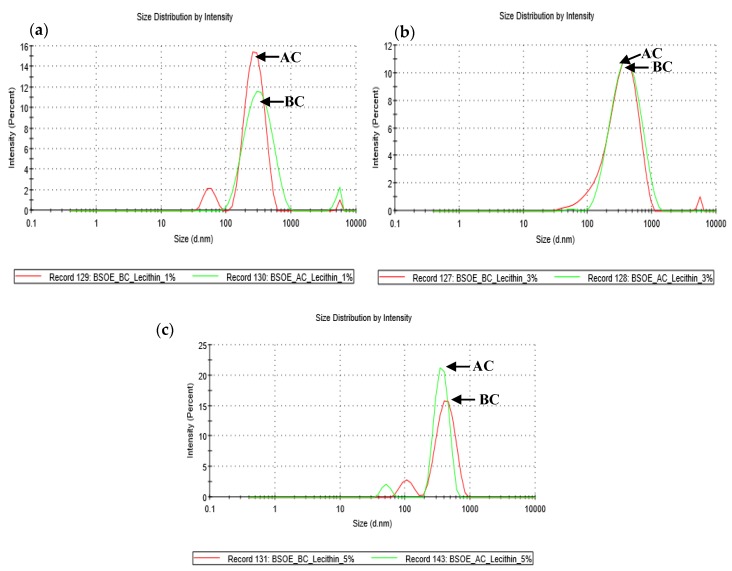
Effect of (**a**) 1% *w*/*v* (**b**) 3% *w*/*v* and (**c**) 5% *w*/*v* lecithin concentrations on particle size distributions of freshly prepared and centrifuged alginate-BSO emulsion. BC = before centrifuge, AC = after centrifuge, *n* = 3.

**Figure 3 pharmaceutics-12-00219-f003:**
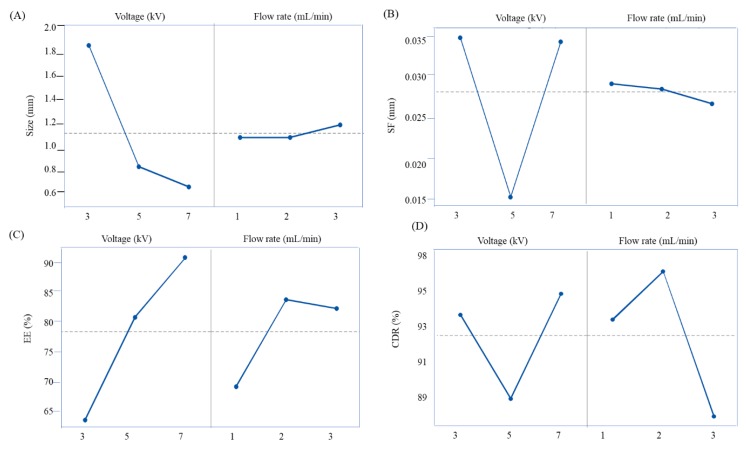
Main effects plot of alginate beads on size (**A**), sphericity factor (SF) (**B**), encapsulation efficiency (EE%) (**C**), % cumulative drug release at 2 h (**D**). Data were presented as mean value (*n* = 3).

**Figure 4 pharmaceutics-12-00219-f004:**
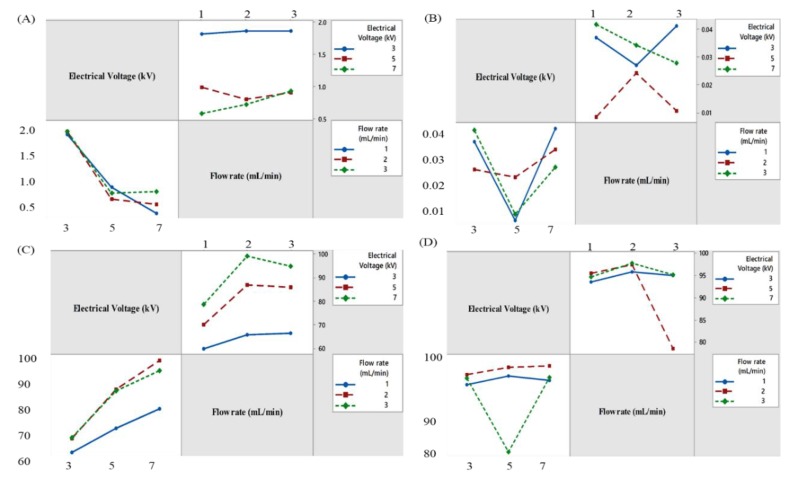
Interaction effect plots of BSO-loaded alginate beads on size (**A**), sphericity factor (SF) (**B**), encapsulation efficiency (EE%) (**C**), % cumulative drug release at 2 h (**D**). Data were presented as mean value (*n* = 3).

**Figure 5 pharmaceutics-12-00219-f005:**
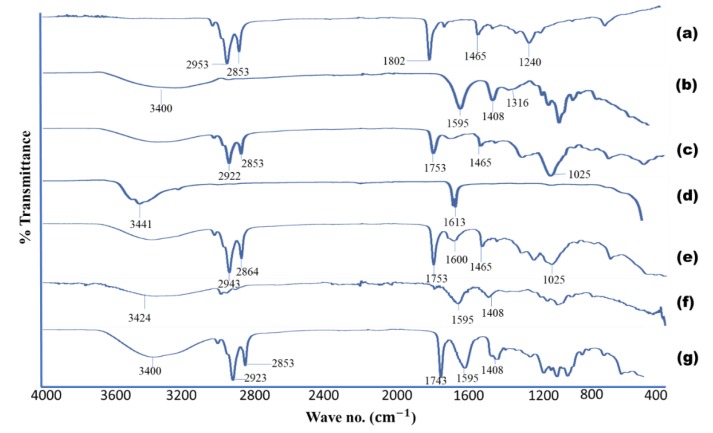
Showing the ATR-FTIR spectra of (**a**) BSO, (**b**) sodium alginate, (**c**) lecithin, (**d**) calcium chloride, (**e**) BSO-free beads, (**f**) alginate-BSO beads, (**g**) physical mixture of all ingredients.

**Figure 6 pharmaceutics-12-00219-f006:**
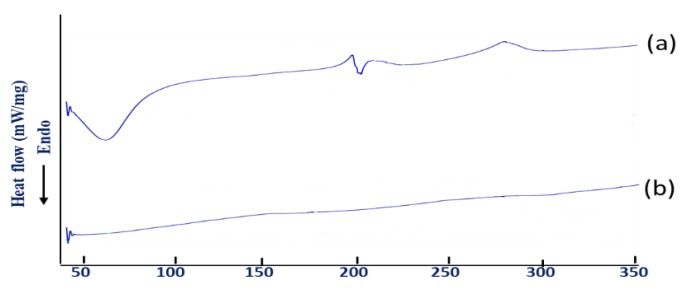
Differential scanning calorimetry (DSC) thermogram of BSO-free beads (**a**) and BSO-loaded beads (**b**).

**Figure 7 pharmaceutics-12-00219-f007:**
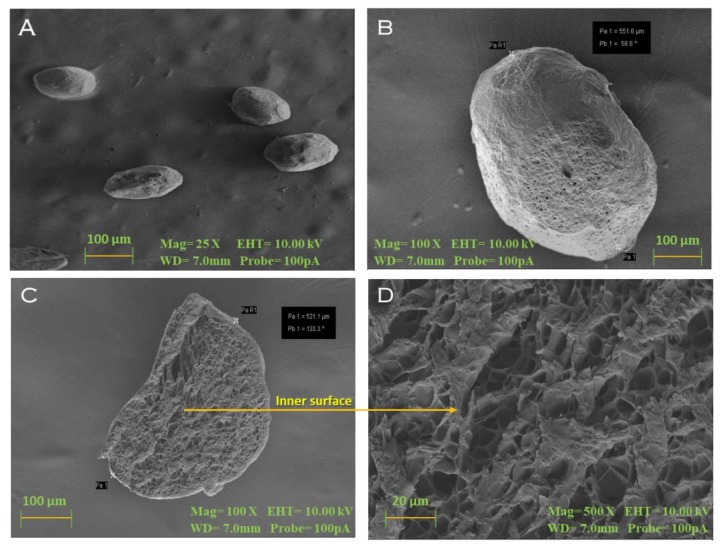
Scanning electron microphotograph of (**A**,**B**) alginate-BSO dried beads showing the surface morphology (25× and 100×) and the presence of pores and channels in the dried beads slice with the magnification of 100× (**C**) and 500× (**D**).

**Figure 8 pharmaceutics-12-00219-f008:**
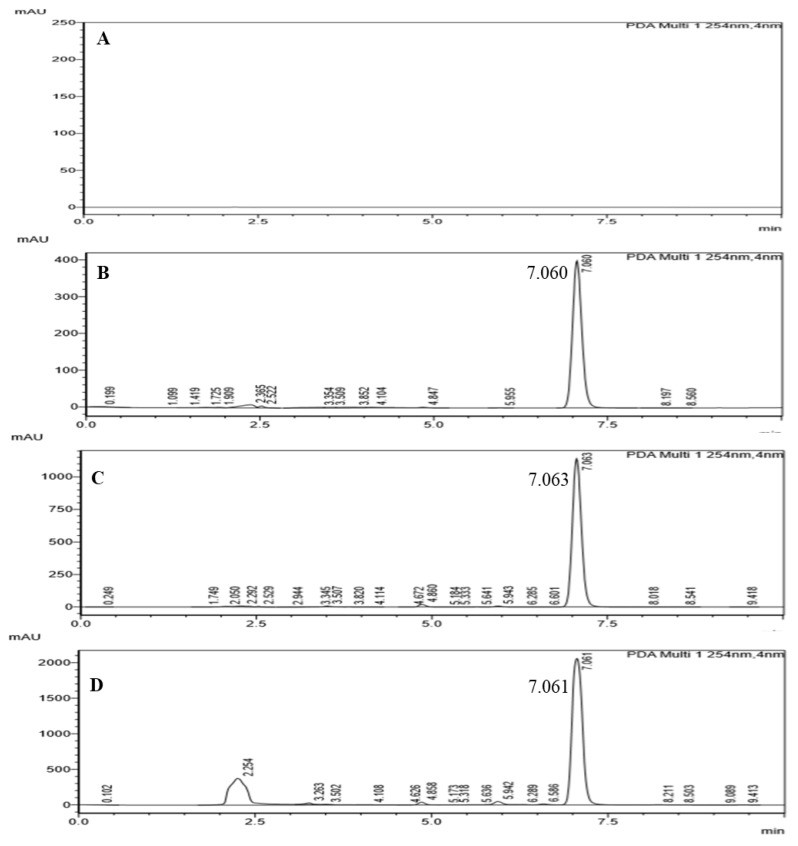
HPLC chromatographs of (**A**) Blank, (**B**) Standard TQ, (**C**) BSO-loaded beads and (**D**) BSO.

**Figure 9 pharmaceutics-12-00219-f009:**
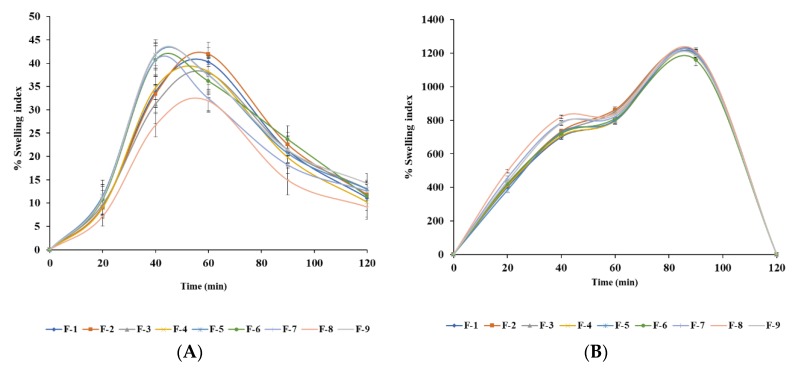
Swelling behaviour of optimized alginate/lecithin beads containing BSO in (**A**) acidic (pH 1.2) and (**B**) buffer (pH 6.8) media. Data were presented as mean ± SD, *n* = 3.

**Figure 10 pharmaceutics-12-00219-f010:**
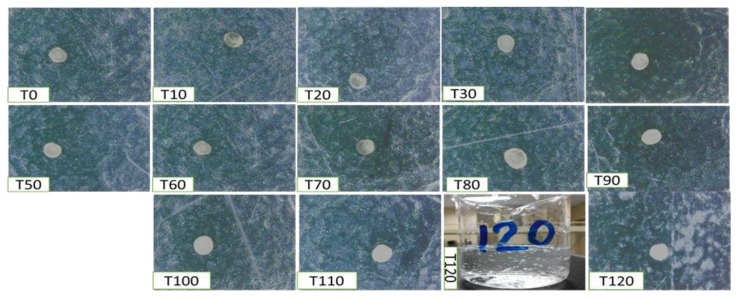
Swollen alginate-BSO bead characterization comprised of evidence on its thickness and graphic appearance in SGF media (pH 1.2) using a digital camera. Magnification at 50×.

**Figure 11 pharmaceutics-12-00219-f011:**
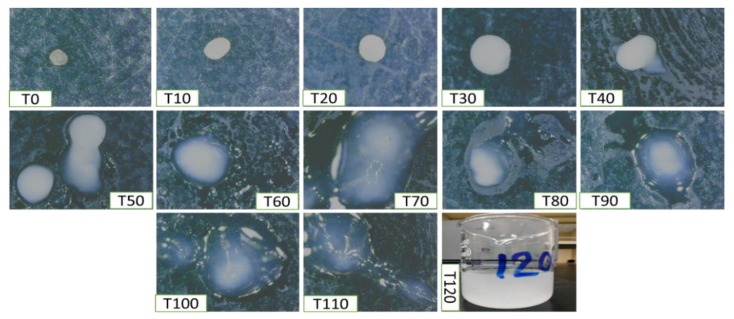
Swollen alginate-BSO bead characterization comprised of evidence on its thickness and graphic appearance in alkaline media (pH 6.8) using a digital camera. Magnification at 50×.

**Figure 12 pharmaceutics-12-00219-f012:**
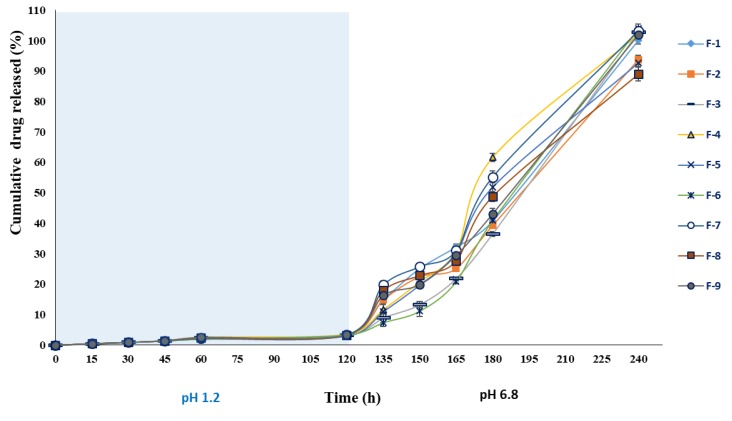
In vitro drug-release profile of alginate-BSO wet beads in simulated gastric fluid (pH 1.2) and simulated intestinal fluid (pH 6.8) (mean ± SD; *n* = 3).

**Figure 13 pharmaceutics-12-00219-f013:**
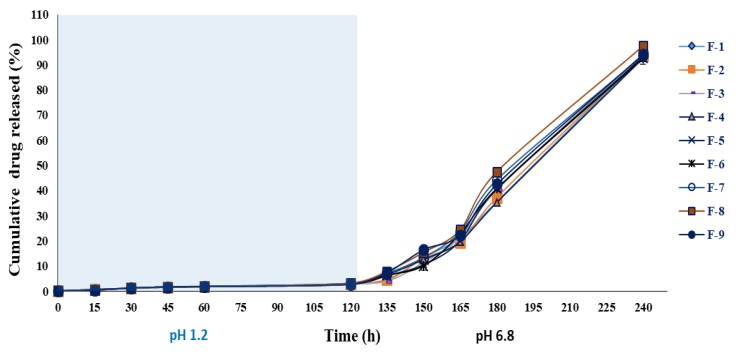
In vitro drug-release profile of alginate-BSO dry beads in simulated gastric fluid (pH 1.2) and simulated intestinal fluid (pH 6.8) (mean ± SD; *n* = 3).

**Figure 14 pharmaceutics-12-00219-f014:**
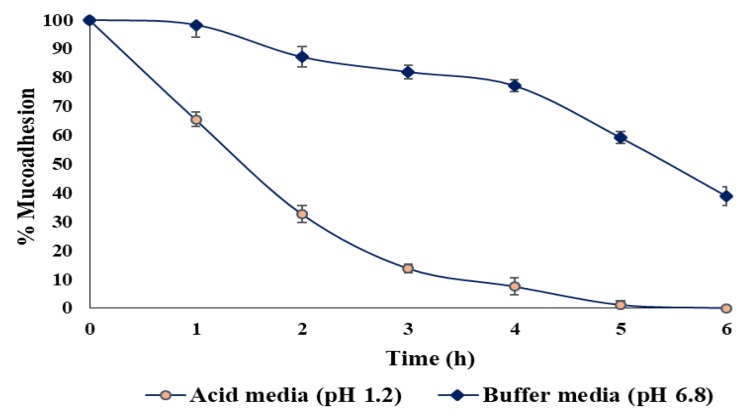
Data of ex vivo wash-off method to evaluate mucoadhesive properties of the optimized alginate beads containing BSO in pH 1.2 and pH 6.8 buffer media (mean ±  SD, *n* = 3).

**Table 1 pharmaceutics-12-00219-t001:** Effect of lecithin concentration on particle size, zeta potential and emulsion stability (ES) of alginate-BSO emulsion. Data were presented as mean ± SD (*n* = 3). * denotes significant difference after centrifugation (*t*-test, *p* < 0.05).

Lecithin Concentration (% *w*/*v*)	Particle Size (nm)		PDI		Zeta Potential (mV)		ES (%)
	Before Centrifugation	After Centrifugation	Before Centrifugation	After Centrifugation	Before Centrifugation	After Centrifugation	
1	253.32 ± 4.6	282.93 ± 14.1 *	0.392 ± 0.0	0.333 ± 0.1	−43.1 ± 0.3	−37.6 ± 2.2 *	92.50 ± 0.2
3	334.51 ± 12.8	347.44 ± 15.1	0.315 ± 0.1	0.358 ± 0.1 *	−46.9 ± 2.2	−43.5 ± 1.3 *	94.83 ± 0.4
5	430.13 ± 14.2	463.23 ± 8.9 *	0.422 ± 0.0	0.548 ± 0.1 *	−51.4 ± 2.7	−45 ± 1.0 *	96.50 ± 0.4

**Table 2 pharmaceutics-12-00219-t002:** Effect of voltage (kV) and flow rate (mL/min) on bead size, sphericity factor and encapsulation efficiency (*n* = 3, mean ± SD).

Code	Voltage (kV)	Flow Rate (mL/min)	Yield (%)	Bead Size (mm)		Sphericity Factor (mm)		Encapsulation Efficiency (%)		Coefficient of Weight Variation (%) = (SD/mean) ×100
				Wet	Dry	Wet	Dry	Wet	Dry	
F1	3	1	90.23 ± 4.19	4.47 ± 0.01	1.82 ± 0.01	0.02 ± 0.00	0.04 ± 0.00	67.20 ± 1.46	59.92 ± 0.90	2.97
F2	3	2	92.11 ± 2.41	4.52 ± 0.03	1.86 ± 0.02	0.01 ± 0.00	0.03 ± 0.00	68.22 ± 3.09	65.89 ± 1.57	2.61
F3	3	3	93.33 ± 3.22	4.76 ± 0.01	1.86 ± 0.01	0.02 ± 0.01	0.04 ± 0.00	70.76 ± 0.86	66.34 ± 1.89	2.43
F4	5	1	91.78 ± 2.65	3.12 ± 0.01	0.99 ± 0.04	0.02 ± 0.01	0.01 ± 0.01	70.11 ± 0.51	70.11 ± 0.51	2.88
F5	5	2	93.33 ± 2.08	2.83 ± 0.05	0.80 ± 0.02	0.02 ± 0.01	0.02 ± 0.02	86.89 ± 1.67	86.89 ± 1.67	3.93
F6	5	3	92.98 ± 3.11	3.30 ± 0.01	0.90 ± 0.02	0.03 ± 0.01	0.01 ± 0.01	86.63 ± 0.83	86.12 ± 0.15	2.97
F7	7	1	93.98 ± 2.84	1.86 ± 0.02	0.72 ± 0.01	0.01 ± 0.00	0.04 ± 0.00	82.16 ± 2.50	78.67 ± 0.90	2.90
F8	7	2	94.87 ± 2.11	1.50 ± 0.02	0.58 ± 0.01	0.02 ± 0.00	0.03 ± 0.00	104.50 ± 4.04	90.13 ± 0.93	1.37
F9	7	3	93.88 ± 4.32	2.58 ± 0.05	0.93 ± 0.01	0.02 ± 0.00	0.03 ± 0.01	96.04 ± 3.61	89.82 ± 154	2.99

**Table 3 pharmaceutics-12-00219-t003:** Swelling parameters (exponent, ns and kinetic constant, ks), swelling rate and water penetration velocity (up to 1 h) of various beads in acidic and alkaline media.

Code	Swelling Exponent (n_s_)		Swelling kinetic Constant (k_s_) (h^−1^)		Correlation Coefficient (r^2^)		Swelling Rate (%/h)		Correlation Coefficient (r^2^)		Water Penetration Velocity (V) (cm/s)	
	pH 1.2	pH 6.8	pH 1.2	pH 6.8	pH 1.2	pH 6.8	pH 1.2	pH 6.8	pH 1.2	pH 6.8	pH 1.2	pH 6.8
F-1	0.74	0.64	12.173	820.55	0.941	0.946	12.15	799.50	0.944	0.981	0.024	1.540
F-2	1.23	0.67	44.47	899.84	0.932	0.976	39.98	876.20	0.887	0.986	0.074	1.533
F-3	1.32	0.72	45.15	827.47	0.928	0.946	40.02	798.54	0.871	0.973	0.027	0.507
F-4	1.49	0.82	53.52	848.16	0.957	0.949	46.76	807.35	0.854	0.959	0.182	2.942
F-5	1.15	0.81	48.54	818.20	0.823	0.964	43.70	781.67	0.835	0.965	0.219	3.897
F-6	1.35	0.58	58.78	814.15	0.917	0.965	51.80	800.48	0.863	0.990	0.082	1.178
F-7	1.38	0.59	59.90	850.94	0.972	0.916	53.39	832.97	0.869	0.977	0.306	4.553
F-8	1.04	0.58	48.05	994.99	0.892	0.988	44.15	982.23	0.900	0.996	0.357	7.620
F-9	1.19	0.66	58.00	930.56	0.982	0.964	53.15	905.95	0.900	0.984	0.163	2.595

**Table 4 pharmaceutics-12-00219-t004:** Dissolution efficiency (DE), mean dissolution time (MDT) and results of the drug release profile fitting to kinetic models of various beads (F-1–F-9).

Code	DE (%)		MDT (min)		Correlation Coefficient (r^2^)										Release Exponent (n)		Gel Characteristic Constant (k_KP_)	
	pH 1.2	pH 6.8	pH 1.2	pH 6.8	Zero order		First order		Higuchi		Hixson crowell		Korsemeyer-Peppas		pH 1.2	pH 6.8	pH 1.2	pH 6.8
					pH 1.2	pH 6.8	pH 1.2	pH 6.8	pH 1.2	pH 6.8	pH 1.2	pH 6.8	pH 1.2	pH 6.8				
F-1	1.10	38.75	54.71	70.29	0.953	0.968	0.954	0.856	0.879	0.757	0.648	0.876	0.763	0.997	0.66	1.47	0.01	0.34
F-2	1.48	38.59	34.88	71.51	0.815	0.958	0.818	0.838	0.964	0.738	0.506	0.884	0.912	0.995	0.49	1.44	0.02	0.33
F-3	1.03	41.19	54.75	67.76	0.937	0.926	0.938	0.867	0.870	0.713	0.711	0.891	0.846	0.909	1.03	1.66	0.01	0.29
F-4	1.82	38.92	38.13	71.29	0.862	0.956	0.864	0.835	0.971	0.738	0.547	0.871	0.958	0.979	0.61	1.29	0.02	0.34
F-5	1.73	47.76	43.06	61.16	0.893	0.928	0.895	0.885	0.947	0.763	0.618	0.838	0.924	0.917	0.88	1.46	0.02	0.38
F-6	1.28	42.13	66.28	68.33	0.969	0.939	0.968	0.840	0.817	0.726	0.759	0.872	0.956	0.900	1.00	1.39	0.01	0.33
F-7	2.22	44.86	43.06	63.31	0.858	0.989	0.861	0.879	0.913	0.833	0.633	0.779	0.851	0.972	1.11	1.02	0.02	0.43
F-8	2.07	43.36	55.06	53.72	0.980	0.951	0.981	0.955	0.896	0.868	0.707	0.701	0.946	0.901	0.88	0.86	0.02	0.41
F-9	1.67	39.97	45.23	69.63	0.860	0.961	0.862	0.840	0.881	0.762	0.652	0.825	0.884	0.962	0.99	1.10	0.02	0.37

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
