# Peer review of "Encapsulation of Black Seed Oil in Alginate Beads as a pH-Sensitive Carrier for Intestine-Targeted Drug Delivery: In Vitro, In Vivo and Ex Vivo Study"

_pharmaceutics, 2020, doi:10.3390/pharmaceutics12030219_

Round 1

Reviewer 1 Report

The authors have presented an interesting study regarding a potential formulation for the study of the therapeutic effect of black see oil.

Although I see no flaws in the technological development of the formulation, with several test confirming the quality of the formulation and the well-designed characteristics the allow for the intended use in vivo, I believe that a better bridge should be made with the intended action in humans.

For this paper to have a good translation potential in regard to therapeutic use in humans a better bibliographic introduction/discussion should be made regarding the reasons for choosing the concentrations of the oil used in the beads. An appropriate bridging study should be cited to justify the use of the concentrations in this study and the future dose to be used in humans with this formulation.

The extensive pharmaceutic/technological study presented here loses impact if the authors does not provide sufficient background on the doses that are plausibly effective in humans.

Release of effective concentrations in humans is mandatory for this formulation to have its innovative aspects.

Therefore, I believe that this bridging information should be added to the introduction and discussion in order for this paper to achieve its maximal interest.

Other than this observation, I believe only a review of the English language (in terms of phrasing construction rather than spelling) is necessary.

If these alterations are performed, I see no objection in the publication of this paper.

Best regards to authors and editors

Author Response

"We would like to ask the authors to change dissolution profile of two pHs in the same figure. Beads behaviour in GIT should be evaluated in sequentially way. Thus, authors are asked to merge figures 12 with 13 and  figures 14 with 15".

Response: Figure have merged according to your suggestion.

Response to reviewer 1

The authors have presented an interesting study regarding a potential formulation for the study of the therapeutic effect of black see oil. Although I see no flaws in the technological development of the formulation, with several test confirming the quality of the formulation and the well-designed characteristics the allow for the intended use in vivo, I believe that a better bridge should be made with the intended action in humans. For this paper to have a good translation potential in regard to therapeutic use in humans a better bibliographic introduction/discussion should be made regarding the reasons for choosing the concentrations of the oil used in the beads. An appropriate bridging study should be cited to justify the use of the concentrations in this study and the future dose to be used in humans with this formulation.

The extensive pharmaceutic/technological study presented here loses impact if the authors does not provide sufficient background on the doses that are plausibly effective in humans. Release of effective concentrations in humans is mandatory for this formulation to have its innovative aspects. Therefore, I believe that this bridging information should be added to the introduction and discussion in order for this paper to achieve its maximal interest.

Other than this observation, I believe only a review of the English language (in terms of phrasing construction rather than spelling) is necessary. If these alterations are performed, I see no objection in the publication of this paper.

Response: bridging information and effective dose related report have added in the introduction section with appropriate justification.

Reviewer 2 Report

General comments

In this manuscript, the authors investigated the feasibility of obtaining the intestinal targeting of back seed oil (BSO) using sodium alginate beads for oral administration. BSO loaded alginate beads were obtained by an electrospray technique and were characterized to determine their morphology and technological properties. The optimized alginate beads containing BSO were orally administered to rats to assess their distribution profile in the stomach and intestine.

Although the topic is interesting, the experimental protocol is neither well described nor well designed, making the scientific soundness of this work questionable.

In addition, English is poor, making some sentences incomprehensible.

Specific comments

The meaning of the abbreviations should be explained the first time they are cited in the text. For instance, at line 42, the meaning of BSO should be explained even if it has already been cited in the abstract.

Line 100. The authors prepared emulsions containing the active ingredient BSO by dispersing BSO (10% w/v relative to the total emulsion volume) in a solution of 1% w/v sodium alginate containing lecithin. The oil phase of these emulsions consisted of the active ingredient BSO only. The authors used a blank emulsion (BSO free alginate beads) as control but they did not explain how they obtained this blank emulsion. What did they use instead of BSO as BSO was the active ingredient and it could not be used in the bank emulsion?

Line 116. The equipment used to perform electrohydrodynamic atomization should be reported. How did the authors separate the alginate beads from the gelling bath? Please, explain.

Line 134. The method used to determine the amount of BSO encapsulated in the beads is unclear. As reported at line 137, the authors measured the absorbance at 600 nm that “represents the turbidity of the resulting emulsion”. How the turbidity of the resulting emulsion was related to BSO content? Please, explain.

Line 222. The procedures used to remove the intestine from the rats should be reported. How long were the beads left in contact with the mucosa surface before washing with PBS? The sentence “Visual inspection was used to record the time needed to detach the beads from rat intestine mucosal surface” is unclear. What is the meaning of “the time needed to detach beads”? How was this time determined? Please, explain.

Line 260. The meaning of the sentence “the reduction in interfacial tension of the oil droplets of the emulsion was restricted by increasing the viscosity of the continuous phase, leading to larger particles” is unclear. Please, rephrase.

Line 267. The authors reported they obtained a good particle size distribution but the plots illustrated in Fig. 2 and the data listed in Table 1 are in contrast to this statement. Polydispersity index (PDI) values were higher than 0.300 ( generally accepted as limit for homogenous particle size distribution) for all formulations, thus indicating a non-homogenous particles size distribution, which is confirmed by the presence of two peaks in Fig. 2 A and C and very broad peak in Fig. 2B. The authors should revise their comments on the particle size distribution according to the data illustrated in Table 1 and Fig. 2.

The authors did not explain clearly which emulsion they used to prepare alginate beads.

In Fig. 2, the abbreviations AC and BC should be explained.

Line 336. The meaning of the sentence “Furthermore, the electrospray and high voltage combination may have potential effect on size, sphericity factor and release profile, as a result this study reported the equalamount of oil release into the targeted site which amount of oil have encapsulated” is unclear. Please, rephrase.

Line 359. Menthol was cited as an ingredient but this molecule was not used in this work. Please, correct.

Line 366. It is unclear how “the stretching vibration bands at 1595.53 and 1408.83 cm-1 maybe due to the interaction with CaCl2” in sodium alginate spectrum. No CaCl2 was present in the neat compound. Please, explain.

Line 408. The following sentences “It can obviously recommend that the alginate matrix beads containing BSO displayed an increase in the thermal peak temperature (Fig. 6 (b)) and enhanced the thermal stability of the BSO loaded beads. It may specify that there were no changes in thermal behaviour of BSO, and it is molecularly dispersed in polymer matrix” are unclear. Please, rephrase. BSO is an oil. What is its thermal behavior? Please, explain.

Line 555. The authors discussed the use of “two different wash-off media” but in the experimental section they described only one was-off medium (phosphate buffer pH 6.8). Please, explain this discrepancy. In addition, the authors reported that data were acquired according to predetermined time intervals from 1 to 6 h but these time intervals were not cited in the experimental section. Please, correct.

English should be carefully revised.

Author Response

Reply to reviewer (2) comments

Reviewer 2

Thanks for your revision and suggestion to make our manuscript in good quality and scientifically sound well. Respected reviewer comments and our scientific response to the reviewers as bellow:

Point 1:

The meaning of the abbreviations should be explained the first time they are cited in the text. For instance, at line 42, the meaning of BSO should be explained even if it has already been cited in the abstract.

Response 1:

The abbreviation of black seed oil (BSO) is now mentioned and explained in the cited text.  The manuscript was thoroughly checked for similar cases.

Point 2:

Line 100. The authors prepared emulsions containing the active ingredient BSO by dispersing BSO (10% w/v relative to the total emulsion volume) in a solution of 1% w/v sodium alginate containing lecithin. The oil phase of these emulsions consisted of the active ingredient BSO only. The authors used a blank emulsion (BSO free alginate beads) as control but they did not explain how they obtained this blank emulsion. What did they use instead of BSO as BSO was the active ingredient and it could not be used in the blank emulsion?

Response 2:

An emulsion containing sodium alginate and lecithin was considered as blank emulsion.

For blank beads the following was added at Line 147:

“BSO-free alginate beads or referred as blank beads were similarly prepared using 1% w/v sodium alginate solution containing lecithin 1, 3 and 5% w/v.”

Point 3:

Line 116. The equipment used to perform electrohydrodynamic atomization should be reported.

How did the authors separate the alginate beads from the gelling bath? Please, explain.

Response 3:

Two explanations were added at Line 134 and Line 141:

“BSO emulsion containing 10% BSO, 1% w/v sodium alginate and 3% w/v lecithin solution was chosen for beads preparation. The emulsion was loaded into 10 mL plastic syringe and pumped through a 22G needle using syringe pump (Shenchen SPLab02, Baoding, China) to control the flow rate. The positive electrode of high voltage power supply (Analog technologies, Inc, San joes, USA) was connected to the needle tip. The collector was grounded beaker filled with 50 mL of 1% w/w calcium chloride (gelling bath).”

“To ensure the complete gelation, the beads were kept in the gelling bath for 30 min for curing with stirring. Then the beads were separated from the gelling bath using stainless steel sieve.”

Point 4:

Line 140. The method used to determine the amount of BSO encapsulated in the beads is unclear. As reported at line 143, the authors measured the absorbance at 600 nm that “represents the turbidity of the resulting emulsion”. How the turbidity of the resulting emulsion was related to BSO content? Please, explain.

Response 4:

Line 167

Point 5:

Line 222. The procedures used to remove the intestine from the rats should be reported. How long were the beads left in contact with the mucosa surface before washing with PBS? The sentence “Visual inspection was used to record the time needed to detach the beads from rat intestine mucosal surface” is unclear. What is the meaning of “the time needed to detach beads”? How was this time determined? Please, explain.

Response 5:

Line 255

Line 261

Point 6:

Line 260. The meaning of the sentence “the reduction in interfacial tension of the oil droplets of the emulsion was restricted by increasing the viscosity of the continuous phase, leading to larger particles” is unclear. Please, rephrase.

Response 6:

 Line 294

Point 7:

Line 267. The authors reported they obtained a good particle size distribution, but the plots illustrated in Fig. 2 and the data listed in Table 1 are in contrast to this statement. Polydispersity index (PDI) values were higher than 0.300   (generally accepted as limit for homogenous particle size distribution) for all formulations, thus indicating a non-homogenous particles size distribution, which is confirmed by the presence of two peaks in Fig. 2 A and C and very broad peak in Fig. 2B. The authors should revise their comments on the particle size distribution according to the data illustrated in Table 1 and Fig. 2.

Response 7:

The statement ‘The emulsions were found to have a good particle size distribution.’ was deleted.

Line 305

Point8:

The authors did not explain clearly which emulsion they used to prepare alginate beads.

Response 8:

Line 141

Point9:

In Fig. 2, the abbreviations AC and BC should be explained.

Response9:

We have corrected and mentioned the abbreviations as BC=before centrifuge, AC=after centrifuge

Point 10:

Line 336. The meaning of the sentence “Furthermore, the electrospray and high voltage combination may have potential effect on size, sphericity factor and release profile, as a result this study reported the equal amount of oil release into the targeted site which amount of oil have encapsulated” is unclear. Please, rephrase.

Response 10:

Line 374

Point 11:

Line 359. Menthol was cited as an ingredient, but this molecule was not used in this work. Please, correct.

Response 11:

Yes, this molecule was not used. It had corrected accordingly.

Point 12:

Line 366. It is unclear how “the stretching vibration bands at 1595.53 and 1408.83 cm-1 maybe due to the interaction with CaCl2” in sodium alginate spectrum. No CaCl2 was present in the neat compound. Please, explain.

Response 12:

Line 412

Several issues were found in the ATR-FTIR discussion, so it was revised and corrected. The band at 1595 in sodium alginate appeared also in the physical mixture but it is not due the interaction. Shifting of this peak to higher value indicated the complex formation.

The figure also was revised due to wrong alignment of the spectra.

Point 13:

Line 408. The following sentences “It can obviously recommend that the alginate matrix beads containing BSO displayed an increase in the thermal peak temperature (Fig. 6 (b)) and enhanced the thermal stability of the BSO loaded beads. It may specify that there were no changes in thermal behaviour of BSO, and it is molecularly dispersed in polymer matrix” are unclear. Please, rephrase. BSO is an oil. What is its thermal behavior? Please, explain.

Response 13:

Line 434

Point 14:

Line 555. The authors discussed the use of “two different wash-off media” but in the experimental section they described only one was-off medium (phosphate buffer pH 6.8). Please, explain this discrepancy. In addition, the authors reported that data were acquired according to predetermined time intervals from 1 to 6 h but these time intervals were not cited in the experimental section. Please, correct.

Response 14:

Line 259

Round 2

Reviewer 2 Report

The authors revised the manuscript properly.